



# Past aridity's effect on carbon mineralization potentials in grassland soils

Zhenjiao Cao[1,2], Yufu Jia[1], Yue Cai[1,2], Xin Wang[1,2], Huifeng Hu[1], Jinbo Zhang[3], Juan Jia[1], Xiaojuan Feng[1,2*]

[1]State Key Laboratory of Vegetation and Environmental Change, Institute of Botany, Chinese Academy of Sciences, Beijing 100093, China

[2]College of Resources and Environment, University of Chinese Academy of Sciences, Beijing 100049, China
[3]School of Geography Sciences, Nanjing Normal University, Nanjing 210023, China

*Correspondence to*: Xiaojuan Feng (xfeng@ibcas.ac.cn)

**Abstract.** Mineralization potential is a key property for assessing carbon substrate's degradability and mineralization in biogeochemical models and studies. While mineralization potential is widely examined under controlled conditions, whether and how it is influenced by the past aridity of sample's origins remain poorly constrained, which is important for an accurate assessment and prediction of future $CO_2$ emissions. Here we collect top- and subsoils from different aridity regimes along a

2100-km grassland transect of northern China and conduct a 91-day decomposition experiment with and without the addition of [13]C-labeled leaf litter under controlled temperature and moisture. $CO_2$ release from both soil organic carbon (SOC) and fresh litter is measured along with microbial biomass, extracellular enzyme activities, soil and mineral properties. We find that neither microbial carbon use efficiency nor biomass-normalized metabolic quotient ($q$CO_2$) is related to the aridity of sampling sites. However, both fresh litter and SOC display the highest mineralization potentials in soils originating from the

driest site. Using pathway analysis, we demonstrate that past aridity's effect is mediated by differential mechanisms for substrates of varied complexity. While microbial biomass plays a more important role in the decomposition of fresh litter, enzyme-catalyzed extracellular reactions predominantly govern the mineralization of SOC. Our findings provide novel evidence on the mechanisms underlying past aridity's effect on the mineralization potentials of organic matter with different qualities, which has significant implications for assessing and modelling decomposition in different aridity regimes.

## 1 Introduction

Organic carbon mineralization is a critical process affecting global carbon and nutrient cycles as well as atmospheric $CO_2$ levels (Wieder et al., 2015). Numerous experiments have demonstrated the primary control of contemporary or experimental climates (including temperature and moisture) on the decomposition of soil organic carbon (SOC) and litter (Davidson and Janssens, 2006; Conant et al., 2011). Recent studies have also underscored the effect of past climate or rainfall patterns on

the contemporary processes of SOC mineralization (Strickland et al., 2015; Hawkes et al., 2017). By comparison, the




influence of past aridity on the mineralization potentials of carbon substrates is less studied. Given that carbon mineralization potentials are commonly measured under controlled temperature and moisture conditions to assess substrate's degradability and potential decay rate (Shaver et al., 2006), it is vital to assess whether and how past aridity of soil's original site affects mineralization potentials in order to fully understand environmental controls on carbon decomposition processes in different

aridity regimes.

Past aridity of sample's origins may influence carbon mineralization potentials via at least four pathways, i.e., through affecting (i) microbial biomass production, (ii) microbial carbon use efficiency (CUE) or metabolic quotient ($q$CO$_2$); (iii) extracellular enzyme activity, and (iv) organic matter and edaphic properties associated with the original soil. The first two microbial responses have been invoked to explain the fast decomposition rate of SOC (after normalizing to SOC content)

from arid regions under similar decomposition conditions (Li and Sarah, 2003; Li and Chen, 2004). Under optimal moisture conditions, microbial communities originating from drier soils show a higher growth rate than those from wetter soils, potentially implying a higher mosiure sensitivity for microbes from arid soils (Li and Sarah, 2003). Moreover, microbes increase energy allocation for respiration under stress including moisture constraint (Odum, 1985) and hence show a higher $q$CO$_2$ or a lower microbial CUE from drier soils (Li and Sarah, 2003). Hence, microbes dwelling in soils of different aridity

regimes may show varied activities under the same incubation conditions (Maestre et al., 2015).

Extracellular enzymes are direct regulators for ex vivo reactions that break organic matter macromolecules into smaller units for the subsequent microbial metabolism (Burns, 1978). The activity and turnover of extracellular enzymes, albeit linked to their microbial producers, are also regulated by abiotic factors, such as temperature, moisture and clay content, etc. (Sinsabaugh, 2010) and hence indicate different facets of microbial processes compared to microbial biomass and $q$CO$_2$

(Sinsabaugh and Follstad Shah, 2012). Recently, enzyme activities are shown to be more sensitive to moisture changes in soils from historically drier than wetter sites (Averill et al., 2016). Hence, it will be important to disentangle mechanisms mediated by extracellular enzymes versus microbial community itself (i.e., biomass production and CUE) that contribute to past aridity's effect on carbon mineralization.

Moreover, past aridity may affect carbon mineralization by adjusting the physiochemical properties of organic matter

from different aridity regimes (Silver and Miya, 2001). In terms of organic matter properties, soils from drier regions are shown to have higher organic carbon to nitrogen (N) ratios (Delgado-Baquerizo et al., 2013) and postulated to show lower SOC mineralization rates under similar conditions compared with those from wetter climates (Marschner and Kalbitz, 2003). Alternatively, soil pH that directly mediates enzyme and microbial activities (Sinsabaugh, 2010) typically increases with increasing aridity (Wang et al., 2014). It is not clear how the above soil properties jointly affect past aridity's effect on

carbon mineralization. Empirical evidence is greatly lacking to disentangle the different mechanisms.

Here we utilize soils collected from grassland sites with varied climatic aridity and conduct soil incubation experiments under controlled temperature and moisture conditions to examine the effect of past aridity (i.e., of soil sampling sites) on carbon mineralization. Given the positive correlation between mineralization rate and SOC concentrations (Harrison-Kirk et





al., 2013), mineralization potential is measured as the percentage of respired $CO_2$ in total organic carbon as a commonly used parameter to assess the degradability of organic matter. Compared to previous studies that employed common litter (Strickland et al., 2015) or reciprocal transplant manipulations (Hawkes et al., 2017), carbon mineralization of soils from different aridity regimes may be further complicated by site-specific edaphic properties such as soil mineralogy and texture (Bronick and Lal, 2005). To control for such side effects, a detailed list of soil properties (including texture, reactive minerals and mineralogy) were examined and compared against mineralization rate. Moreover, we add $^{13}C$-labeled leaf litter to soils collected from different depths to examine litter mineralization without inducing site-specific edaphic properties and to compare the mineralization of organic matter with varied complexity (i.e., fresh litter, top- and subsoils). Finally, using pathway analysis coupled with measurements of microbial communities, extracellular enzyme activities and soil properties, we attempt to quantitatively assess mechanisms contributing to the effect of past aridity on mineralization potentials. Specifically, we hypothesize that past aridity mediates SOC mineralization via its effect on microbial, enzyme and soil organic matter (SOM) properties (Hypothesis 1). Moreover, given the vital role of extracellular enzymes in macromolecule breakdown within complex soil matrix, extracellular enzymes have a stronger influence on regulating aridity's effect on the mineralization of SOC relative to fresh litter (Hypothesis 2).

## 2 Materials and Methods

### 2.1 Study area and soil sampling

Six grassland sites with varied climatic conditions are selected from a 2100-km transect from northern China (37.03˚N–47.60˚N; 98.67˚E–119.50˚E; 1060–3613 m above the sea level; Fig. S1). These sites represent typical grasslands with minimal human influences, including three alpine sites on the Qinghai-Tibetan Plateau (Halihatu, HLHT; Daqiao, DQ and Haibei, HB) and three temperate sites in Inner Mongolia (OtogBanner, OB; XilinGol League, XG and Xilin Hot, XH). Mean annual temperature (MAT) ranges from –1.8 to 6.4˚C, and mean annual precipitation (MAP) varies from 256 to 422 mm (WorldClim database; http://www.worldclim.org; Table 1). Aridity index, calculated as the ratio of precipitation to potential evapotranspiration, is used to indicate regional dryness of the sampling sites and ranges from 0.28 to 0.61 (http://www.cgiar-csi.org/). Soil types include Arenosols, Kastanozems Chernozems and Cambisols (IUSS Working Group WRB; 2006) and the dominant vegetation types are listed in Table 1. Parent materials are dominated by clastic and igneous rocks and the dominant minerals are similar across sites including quartz and feldspar besides small amounts of calcite (Table S1).

Three random soil cores (up to 100 m in between) were taken from each site in July-August 2015. Soils from two horizons/depths were collected: topsoil (0–10 cm) from the A horizon and subsoil (50–70 cm for HLHT and DQ; 30–40 cm for HB; 30–50 cm for the other sites) from the B horizon. The samples were separated into two parts: one part was air-dried immediately for soil physiochemical analyses and the other part was stored in ziplock bags in the dark at 4˚C for the



incubation experiment. Both parts were passed through a 2-mm sieve with visible roots removed and homogenized before further treatment. Soils from different cores were not mixed and hence represented authentic field replicates.

## 2.2 Analysis of soil properties

Total carbon and N contents were measured for the air-dried soils using an elemental analyzer (Vario EL III, Elementar,
Hanau, Germany). SOC content was calculated by subtracting inorganic carbon from total carbon, with the former analyzed volumetrically by reaction with hydrochloric acid (HCl). Soil pH was analyzed by a pH meter in a soil:water suspension (1:2.5, w:v). Water-extractable organic carbon (WEOC) was extracted from air-dried original soils (~5 g) by mixing with 12 mL Milli-Q water on a reciprocal shaker for 24 h. The supernatant was filtered through 0.45-μm PTFE filters after centrifugation and acidified to pH < 2 with HCl for WEOC measurement on a Multi N/C 3100 total organic carbon analyzer
(Analytik Jena, Germany). Soil texture was examined using Malvern Mastersizer 2000 particle analyzer after removing SOM and calcium carbonates (Ma et al., 2018). Reactive iron ($Fe_d$) and aluminum ($Al_d$) were extracted by the citrate-bicarbonate-dithionite method (Lalonde et al., 2012), and their contents were determined on an inductively coupled plasma-atomic emission spectrometer (ICP-AES; ICAP 6300, Thermo Scientific, USA).

## 2.3 Incubation experiment

Within one month of sample collection, authentic field replicates of soils were incubated in the dark at 25°C for 91 days to examine carbon mineralization. Varying amounts of soils were weighed into 165-ml brown glass flasks containing >60 mg of SOC. Soil water content was maintained at 55–60% of the water holding capacity by regularly weighing and spraying MilliQ water over the soil. Before incubation, all samples were pre-incubated under the same condition for two weeks to activate soil microbes. On the first day of incubation, one half of the replicates was used as control without any amendment,
while the other half was mixed with fine powders (~ 2500 mesh) of $^{13}C$-labelled grass leaves (mixture of *Oplismentls undulatifolius folius* and *Miscanthus sinensis*) to examine the mineralization of fresh litter relative to SOC. Due to logistic reasons, $\delta^{13}C$ of the added leaves varied between HB (2067.75‰ in the first batch of incubation) and all other soils (1269.97‰ in the second batch). Nonetheless, both $\delta^{13}C$ values were substantially higher than those of SOC (–26.35 to –23.06‰) and did not influence the calculation. The added litter carbon corresponded to a higher proportion of SOC in the HB (0.7%) than
all other soils (0.29%) due to an oversight in calculation. Nevertheless, the low amendment rates did not induce priming effect on the mineralization of native SOC in any soil (details in Results).

Mineralization was monitored by quantifying $CO_2$ accumulated in the headspace for 6 h on >12 selected days using gas chromatograph (GC; Agilent 7890A, USA) coupled with a flame ionization detector (FID). To differentiate SOC- and litter-derived $CO_2$, $\delta^{13}C$ of $CO_2$ from litter-amended samples was measured periodically (5-6 times in total) on an isotope ratio
mass spectrometry (IRMS; Delta PLUS XP, Thermo Finnigan, Germany). The contribution of SOC- and litter-derived carbon to $CO_2$ was calculated by the mass balance equations:




$$r_t = r_{SOC} + r_{litter} \qquad (1)$$

$$r_t \times \delta^{13}C_t = r_{SOC} \times \delta^{13}C_{SOC} + r_{litter} \times \delta^{13}C_{litter} \qquad (2)$$

where $r$ is cumulative $CO_2$ (mg C g$^{-1}$ soil), the subscript $t$ refers to total respired $CO_2$ from the litter-amended sample. The mineralization potential for litter ($R_{litter}$) as well as SOC in the control ($R_{control}$) and litter-amended treatments ($R_{SOC}$) was normalized to the corresponding organic carbon content. Microbial metabolic quotient ($qCO_2$) was calculated as cumulative $CO_2$ divided by microbial biomass (estimated using phospholipid fatty acids; PLFAs; Section 2.4) at the end of incubation (Martínez-García, et al., 2018).

### 2.4 Analyses of PLFAs and extracellular enzyme activity

Microbial community structure and biomass were analyzed by PLFAs using a modified Bligh-Dyer extraction (Bligh and Dyer, 1959) at the end of the incubation (details in Supplementary Materials and Methods 1.1). PLFAs are categorized into non-specific, fungi-, Gram-positive (G+) and Gram-negative (G–) bacteria-derived (Harwood and Russell, 1984). The concentration of individual PLFAs was normalized to the SOC content. Microbial community composition is assessed by the ratio of fungal to bacterial PLFAs (F/B) and the ratio of G+ to G– bacteria (G+/G–). The $\delta^{13}C$ values of individual PLFAs analyzed on gas chromatography coupled to a stable isotope ratio mass spectrometry via a combustion interface (GC-C-IRMS) and the proportion of litter-derived carbon in PLFAs was calculated using a mass balance approach (Supplementary Materials and Methods 1.1).

Microbial CUE was calculated as below (Kallenbach et al., 2016):

$$CUE = \frac{PLFA-C_{litter}}{PLFA-C_{litter}+CO_2-C_{litter}} \times 100\% \qquad (3)$$

where PLFA-$C_{litter}$ and $CO_2$-$C_{litter}$ are the amount of carbon in litter-derived PLFAs and $CO_2$, respectively (Bradford et al., 2013).

At the end of the incubation, the activity of one oxidase (phenol oxidase) and four hydrolases including α-glucosidase, β-glucosidase, alkaline phosphatase and leucine-aminopeptidase were measured according to Saiya-Cork et al. (2002; Supplementary Materials and Methods 1.2). Enzyme activity was expressed as specific activity normalized to the SOC content (Allison et al., 2014).

### 2.5 Data and statistical analysis

We assessed the homogeneity of variances and normal distribution of data using Shapiro-Wilk test before applying parametric methods. Non-parametric tests (Kruskal-Wallis H or Wilcoxon) were conducted for non-normally distributed data. Differences of measured soil and microbial properties in soils from different depths or treatments were assessed by paired T or Wilcoxon test. Relationships between environmental factors and substrate mineralization potentials were assessed using Pearson (for normally distributed data) or otherwise Spearman correlations by IBM SPSS 20.0 (IBM SPSS, Chicago, USA). Differences and correlations were considered to be significant at a level of $p < 0.05$.





To delineate mechanisms regulating substrate mineralization potentials, structural equation modelling (SEM) was conducted to quantify the complex interactions between environmental variables and substrate mineralization potential using the 'lavaan' package of R software (version 3.5.3; Rosseel, 2012). The selection of model parameters and optimization of the model were detailed in Supplementary Materials and Methods 1.3. To complement SEM in evaluating the main influencing variable(s) on mineralization potentials, multiple stepwise regression was conducted encompassing the same variables (aridity index, soil minerals, pH, SOM property, PLFAs, phenol oxidase and hydrolases) using IBM SPSS 20.0 (IBM SPSS, Chicago, USA) at a level of $p < 0.05$. Variables not contributing to the variation of substrate mineralization potential were excluded in the subsequent variable selection process based on the $p$ values (i.e., only variables with $p \leq 0.10$ were retained in the model). After the establishment of regression models, normal distribution of model residues was checked while collinearity among the selected variables was avoided based on the variance inflation factor (a cutoff value of 1 was chosen to define collinearity) to ensure robustness of the models. The explanatory power of the regression model is indicated by $R^2$.

# 3 Results

## 3.1 Soil bulk properties

Subsoils had higher pHs, SOC and N contents than their corresponding topsoil ($p < 0.05$) but similar SOC/N ratios ($p > 0.05$; Table 2). Concentrations of WEOC were higher in the subsoil than the topsoil after normalization against SOC ($p < 0.05$). The driest site OB showed the highest pH value, the lowest SOC and N contents and SOC/N ratios while the wettest site HB showed the highest SOC, N contents and SOC/N ratios ($p < 0.05$). Although the SOC-normalized concentration of WEOC was highest in the topsoil of OB, it was not related to the aridity index for either top- or subsoils. Contents of $Fe_d$ and $Al_d$ did not show consistent patterns related to aridity or depth. Clay content was lowest for both top- and subsoils in OB ($p < 0.05$).

## 3.2 Microbial PLFAs and $\delta^{13}C$

The top- and subsoils from the same site had similar concentration of total PLFAs, F/B and G+/G– ratios ($p > 0.05$; Table 3). The F/B ratio was highest in OB at both depths ($p < 0.05$). Litter amendment did not produce significant effects on either PLFA concentrations or ratios ($p > 0.05$). PLFA concentrations were negatively correlated with aridity index in both treatments ($p < 0.05$; Fig. S2). The $\delta^{13}C$ of PLFAs was analyzed for all the sites except HB due to sample loss. There were no significant differences in $\delta^{13}C$ of PLFAs for various microbial groups in either treatment ($p > 0.05$; Fig. S3). The abundance-weighted average $\delta^{13}C$ of PLFAs was higher in the topsoil of HLHT than the other sites ($p < 0.05$) and was similar in all subsoils ($p > 0.05$). The proportion of litter-derived C in PLFAs was around 1–3% in the topsoil and 1–5% in the subsoil (Table S2), showing no difference between depths ($p > 0.05$).

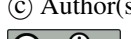



### 3.3 Mineralization, $q\text{CO}_2$ and CUE

$R_{control}$ was 1.4–10.6% and 3.0–14.2% in the top- and subsoils (Fig. 1), similar to $R_{SOC}$ in the litter-amended soils ($p > 0.05$). Both were highest in the driest OB soil with the lowest SOC contents. By comparison, $R_{litter}$ was higher (17.9–95.5% and 35.6–92.3% in the top- and subsoils, respectively; Fig. 1) and also highest in OB for both depths. All mineralization

potentials were negatively correlated with aridity index ($p < 0.05$; Fig. 2). Similar $q\text{CO}_2$ values were found for the litter-amended and control treatments ($p > 0.05$; Table 3). There were no significant differences between top- and subsoils except for OB and DQ. Microbial CUE did not significantly differ between depths ($p > 0.05$; Table 3). Neither $q\text{CO}_2$ nor CUE showed a relationship with aridity index ($p > 0.05$; Fig. S2).

### 3.4 Extracellular enzyme activity

The SOC-normalized specific activity of hydrolases was highly variables across sites while phenol oxidase activity was highest in OB at both depths (Table 4). Litter amendment did not produce any effect on enzyme activities at either depth ($p > 0.05$). Enzyme activity did not show consistent variations between depths. Phenol oxidase and leucine-aminopeptidase activities were negatively correlated with aridity index in both treatments ($p < 0.05$; Fig. S2).

### 3.5 Relationships between mineralization potentials and environmental variables

All mineralization potentials are negatively correlated with aridity index and positively correlated with soil pH and phenol oxidase activity ($p < 0.05$; Fig. 2). $R_{control}$ and $R_{SOC}$ also show positive correlations with WEOC concentrations and leucine-aminopeptidase activities and negative correlations with SOC contents and SOC/N ratios ($p < 0.05$). $R_{control}$ is positively correlated with the F/B ratio despite a low r value (r = 0.36; $p < 0.05$). By comparison, $R_{litter}$ is strongly and positively correlated with total PLFAs (r = 0.60; $p < 0.05$). This relationship is confirmed by positive correlations between $R_{litter}$ and

various PLFA groups (i.e., fungal, G+ and G– bacterial PLFAs; $p < 0.05$) while no significant correlations were observed for either $R_{control}$ (except with fungal PLFAs) or $R_{SOC}$ (Fig. S4). Soil minerals, CUE and G+/G– ratio hardly show correlations with substrate mineralization potentials.

To disentangle the interactive mechanisms, we conducted an SEM analysis for the pooled data from both soil depths. Combining top- and subsoils allowed us to focus on the comparison of pathways affecting the mineralization of SOC versus

litter and also improved the model performance by increasing the number of data. The constructed SEMs show a good model fit indicated by a non-significant $\chi^2$ test ($p > 0.05$), a high comparative fit index (CFI > 0.95), a low root mean square error of approximation (RMSEA < 0.05) and a bootstrap $p$ value ($p > 0.1$; Schermelleh-Engel and Moosbrugger, 2003), and explains 89%, 79% and 41% of variations in $R_{control}$, $R_{SOC}$ and $R_{litter}$, respectively. Based on the SEMs, enzyme activities are the most important direct regulator for SOC mineralization in both control and litter-amended treatments (Figs 3a-b). By contrast,

microbial biomass (PLFAs) displays a stronger direct influence on $R_{litter}$ than either hydrolyses or phenol oxidase (Fig. 3c). Phenol oxidase instead of hydrolases affects SOC mineralization potentials in both treatments while hydrolases have a



similar effect on $R_{litter}$ as phenol oxidase. Soil minerals and SOM property have a relatively minor, negative effect on all mineralization potentials via influencing enzyme activities and PLFAs. In addition, soil pH has a positive effect on all mineralization potentials through positively influencing PO activity. Aridity index has an overall negative effect on all mineralization potentials via exerting a positive effect on SOM property and a negative effect on soil pH and PLFAs. It is

also notable that hydrolases are influenced by soil minerals and SOM property only in the litter-amended treatment. Generally, these pathways are consistent with correlations between the corresponding variables (Fig. S2) and mineralization potentials (Fig. 2).

To verify the differential impacts of PLFAs and enzyme activities on mineralization potentials, we further employed multiple stepwise regression encompassing the same environmental variables. The standardized partial regression coefficient

is used to assess the relative importance of influencing factors, i.e., a higher value indicates a stronger influence. The regression analysis yields $R^2$ of 0.85, 0.76 and 0.42, indicating a reasonable explanatory power of the model (Table 5). Consistent with the SEM results, phenol oxidase activity is the most important variable influencing SOC mineralization in both treatments. By comparison, PLFAs exert the strongest control on $R_{litter}$.

## 4 Discussion

### 4.1 Magnitude of mineralization potentials for litter versus SOC

Mineralization potential of SOC ranged from 1.4% to 14.2% in both control and litter-amended treatments in our studied grasslands, falling within the range reported for other grassland soils (Guenet et al., 2010; reference details in Table S3). By comparison, the mineralization potential of grass leaf litter ($R_{litter}$) showed a wider range and higher values in this study (17.9–95.5%) compared to the literature (Sievers and Cook, 2018; Table S4). As shown in Fig. 2 and discussed below, the

mineralization potential of the same litter increases with increasing aridity of the original site. Sites in this study had an aridity index of 0.28–0.61, much dried than those in the above studies. Hence, the high values of $R_{litter}$ in our study may also reflect the high mineralization potential of litter in semiarid regions. The mineralization potential is significantly higher for grass leaf litter than for SOC ($p < 0.05$), reflecting the high degradability of fresh litter carbon due to microbial preference for litter enriched with labile carbon such as carbohydrates (Guenet et al., 2010) and the absence of mineral protection

compared to SOC (Six et al., 2002).

### 4.2 Differential controls by extracellular enzymes and microbial biomass on the mineralization of SOC versus fresh litter

An important finding of this study is that while $R_{litter}$ is mainly affected by the SOC-normalized concentrations of PLFAs (i.e., microbial biomass) in the soil, extracellular enzyme (specifically, phenol oxidase) activities rather than PLFAs

predominantly govern the mineralization of SOC. This result is supported by both SEM (Fig. 3) and multiple stepwise





regression analyses (Table 5), in line with our second hypothesis. The production of extracellular enzymes is not only related to the size of microbial community (biomass) but also to its structure (Gallo et al., 2004). As >95% of microbial biomass is considered to be dormant or inactive in the soil (Fierer, 2017), these microbes may not actively participate in enzyme production. Moreover, extracellular enzymes may experience inactivation or prolonged activity caused by sorption to
minerals and/or complexation with SOM (Arnosti et al., 2014). Hence, microbial biomass and extracellular enzymes may act quite independently on carbon decomposition.

As SOM consists of a consortium of complex molecules, often in association with each other and/or minerals (Lehmann and Kleber, 2015), its decomposition is a multi-step process initiated with the oxidation or hydrolysis by extracellular enzymes (Nannipieri et al., 2002). As such, macromolecular SOM structures are broken into molecules small enough to be
transported through microbial cell membranes and utilized for respiration and biomass production (Sollins et al., 1996). Hence, enzymatic depolymerisation is a crucial, rate limiting step for large, complex substrates (Conant et al., 2011). This explains the dominant role of extracellular enzyme activity in SOC mineralization in both treatments in our experiment. In contrast, for leaf litter that is relatively easy to degrade without complex mineral interactions (Bosatta and Ågren, 1999), microbial biomass predominantly controls its mineralization.

Interestingly, among the investigated enzymes, phenol oxidase rather than hydrolyses plays a decisive role in SOC mineralization in our study, likely related to the much higher activity of phenol oxidase compared with hydrolases in the soil (Table 4). It also agrees with previous findings that oxidases (i.e., phenol oxidase and catalase) instead of hydrolases (i.e., urease and neutral phosphatase) are key players in SOM breakdown, controlling its decomposition rate (Hassan et al., 2013). Oxidative enzymes are shown to be more important in the soils of desert grasslands than temperate grasslands (Stursova and
Sinsabaugh, 2008). Phenol oxidase activity is also documented to control SOC decomposition and $CO_2$ emission in peatlands and some upland ecosystems (Freeman et al., 2001). By comparison, in contrast to SOC, hydrolases are more important than phenol oxidase in the mineralization of litter that contains more hydrolyzable carbon such as cellulose (Fig. 3).

### 4.3 Pathways regulating past aridity's effect on carbon mineralization potentials

Our study demonstrates that the aridity index of sampling sites has a strong negative effect on the mineralization potential of
both SOC and litter added to the soil (Fig. 2), such that the direst site OB displays the highest mineralization potential for both substrates under controlled temperature and moisture conditions. This finding underscores past climate's effect on carbon mineralization potentials and agrees with other reports showing elevated soil respiration (after normalizing to SOC content) and/or microbial metabolic quotient in soils from arid than sub-humid regions under similar incubation conditions (Li and Sarah, 2003). More importantly, employing SEM analysis (Fig. 3), we show that, consistent with our first hypothesis,
the relationship between past aridity and substrate mineralization potential is jointly mediated through aridity's effect on microbial properties (i.e., phenol oxidase activity and PLFAs) and SOM property (for mineralization of SOC only).





First, soils from drier regions (with a lower aridity index) exhibited higher phenol oxidase activity under incubation conditions (Fig. S2). This result agrees with the higher responsiveness of enzyme activities to water availability in drier soils (Averill et al., 2016), which is considered to reflect microbial strategies to cope with sporadic supply of water in arid environments. Aridity index also influences phenol oxidase activity via a negative effect on soil pH and a positive effect on

SOM property (affected by SOC contents and SOC/N ratios; Fig. S2c). The former pathway is likely related to pH's positve effect on phenol oxidase activity in neutral-to-alkaline soils (Sinsabaugh, 2010). The latter pathway is attributed to increasing phenol oxidase activity with both decreasing SOC/N ratios (Artigas et al., 2008) and increasing WEOC concentrations (Fang et al., 2015). With the above pathways combined, aridity index and pH indirectly exert a negative and positive effect on substrate mineralization potentials, respectively, with the latter relationship in accordance with previous

reports (Whittinghill and Hobbie, 2011; Carrasco et al., 2017).

In contrast to phenol oxidase, hydrolases are unresponsive to the variation of aridity index or any other investigated variables in this study. With a lower activation energy ($E_a$), phenol oxidase ($E_a$ of 32.5 kJ mol$^{-1}$) is often more active than hydrolases (e.g., β-glucosidase: $E_a$ of 61.8 kJ mol$^{-1}$; Davidson et al., 2012) and less stable in the environment (Sinsabaugh, 2010). Additionally, cellulolytic activity shows small variations while phenol oxidase activity typically exhibits a large

decline in decaying organic matter (Carreiro et al., 2000). Hence, phenol oxidase is more responsive to environmental variabilities than hydrolases and its activity is strongly linked to substrate mineralization potentials, especially for SOC (Fig. 3).

Second, in contrast to previous studies (Odum, 1985; Li and Sarah, 2003), neither $q\mathrm{CO}_2$ nor CUE in our experiment shows any consistent changes with shifting aridity of the original sites. Hence, aridity index mediates $R_{litter}$ mainly via

negatively influencing PLFA concentrations (Fig. 3c). Similar to phenol oxidase activities, microbial growth in soils from drier regions may be strongly promoted during the incubation due to the release of moisture constraint (Li and Sarah, 2003), which in turn leads to a higher mineralization rate of the easy-to-degrade carbon (e.g., leaf litter).

Third, past aridity also affects SOM property that indirectly regulates mineralization potentials of both SOC and litter via affecting phenol oxidase and hydrolases (for $R_{litter}$ only). In this study, SOM property is an arbitrary term generated by

PCA for SOC contents, SOC/N ratios and WEOC concentrations (Table S5). Its negative effect on SOC mineralization potentials is related to SOC and SOC/N's negative correlations and WEOC's positive correlation with $R_{control}$ and $R_{SOC}$ (Fig. 2), in agreement with the literature data (Fig. S5). In addition, high SOC/N ratios and low WEOC concentrations may lower SOC mineralization potentials due to N and/or labile carbon constraints on microbial activities (Kalbitz et al., 2000; Schimel and Weintraub, 2003).

Last but not least, soil minerals that are strongly influenced by parent materials rather than aridity (Harradine and Jenny, 1958) may exert complicating effects on SOC mineralization potentials via interacting with SOC (von Lützow et al., 2006) and microbial activities (Bruun et al., 2010). In our studied soils, mineralization potentials are not directly correlated with $Fe_d$, $Al_d$ or clay contents ($p > 0.05$). However, soil minerals have indirect effects on mineralization potentials via positively

affecting SOM property (mainly SOC and N contents) as well as hydrolases (in the litter-amended treatments) and negatively affecting PLFAs in the SEM. The former two relationships reflect minerals' protective effect on SOC (von Lützow et al., 2006) and extracellular enzymes (Wei et al., 2014) while the latter may be associated with the inhibitive effects of reactive Fe and Al on microbial respiration and growth (Bruun et al., 2010; Lemire et al., 2013). Nonetheless, soil

minerals have a minimal added effect on $R_{control}$ and $R_{SOC}$ and a relatively minor effect on $R_{litter}$ compared to other aridity-influenced variables. We hence conclude that past aridity of the sampling sites has a strong control on carbon mineralization potentials of both SOC and litter mainly via mediating microbial biomass, enzyme activities and SOM property.

## 5 Conclusion

In summary, our study demonstrates that the aridity of sampling sites has a strong and consistent effect on the mineralization

of both common litter and SOC from grasslands under controlled conditions. Such effects should be taken into account in the assessment of carbon release potentials, given the wide application of controlled incubation in studying carbon mineralization. Moreover, in comparison with the well-investigated microbial control on climate's legacy effect (Strickland et al., 2015; Hawkes et al., 2017), our study emphasizes the importance of extracellular processes catalyzed by enzymes (in particular, phenol oxidase) in the mineralization of more complex SOC relative to fresh litter. As extracellular enzyme and

microbial activities may show varied responses to climatic variations, our findings suggest different vulnerabilities for organic matter of different qualities and originating from various aridity regimes. With aridity shifts in the future, soil carbon stocks in drylands may be more vulnerable to decomposition than those in humid regions.

## Author contribution

XF designed the study; ZC and YC carried out the experiment and analyses with help from JJ; HH, JZ, YC and XW

collected soil samples; ZC analyzed the data with help from YJ and JJ; ZC, YJ and XF wrote the manuscript with input from all the other authors. ZC and YJ contributed equally to this work.

## Acknowledgements

This study was supported financially by the Chinese National Key Development Program for Basic Research (2017YFC0503902, 2015CB954201), the National Natural Science Foundation of China (41422304, 41773067, 41807329)

and the International Partnership Program of Chinese Academy of Sciences (Grant No. 151111KYSB20160014).



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





**Table 1. Location and basic information of the studied grassland sites.**

| Site | Latitude (˚N) | Longitude (˚E) | Altitude (m) | MAT (˚C) | MAP (mm) | Aridity index | Soil type | Vegetation type |
|------|------|------|------|------|------|------|------|------|
| OB | 39.15 | 107.93 | 1532 | 6.41 | 272 | 0.28 | Arenosols | *Stipa breviflora* |
| DQ | 37.13 | 99.49 | 3252 | 0.67 | 271 | 0.36 | Kastanozems | *Achnatherum splendens* |
| HLHT | 37.03 | 98.67 | 3613 | –0.71 | 256 | 0.36 | Kastanozems | *Stipa purpurea* |
| XH | 43.58 | 116.69 | 1231 | 1.30 | 343 | 0.42 | Kastanozems | *Cleistogenes squarrosa* |
| XG | 46.60 | 119.50 | 1060 | –1.76 | 411 | 0.56 | Chernozems | *Stipa baicalensis* |
| HB | 37.60 | 101.32 | 3258 | –0.22 | 422 | 0.61 | Cambisols | *Kobresia humilis* |

MAT: mean annual temperature; MAP: mean annual precipitation; aridity index is defined as the ratio of MAP to potential evapotranspiration and increases with increasing moisture; OB: OtogBanner, DQ: Daqiao, HLHT: Halihatu, XH: Xilin Hot, XG: XilinGol League, and HB: Haibei.

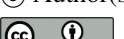


**Table 2. Bulk properties of the studied soils (mean ± standard error; n = 3)**

| Site | Soil pH | SOC (%) | N (%) | SOC/N | WEOC (mg g$^{-1}$ SOC) | Fe$_d$ (mg g$^{-1}$ soil) | Al$_d$ (mg g$^{-1}$ soil) | Clay (%) |
|---|---|---|---|---|---|---|---|---|
| *Topsoil* | | | | | | | | |
| OB | 9.22 ± 0.02 | 0.35 ± 0.01 | 0.05 ± 0.00 | 6.66 ± 0.24 | 10.74 ± 4.87 | 2.49 ± 0.08 | 0.23 ± 0.01 | 0.48 ± 0.09 |
| DQ | 8.29 ± 0.05 | 2.32 ± 0.19 | 0.28 ± 0.02 | 8.43 ± 0.07 | 2.84 ± 0.06 | 5.61 ± 0.12 | 0.59 ± 0.02 | 3.73 ± 0.14 |
| HLHT | 8.05 ± 0.07 | 4.68 ± 0.65 | 0.46 ± 0.06 | 10.31 ± 0.38 | 4.11 ± 0.24 | 7.00 ± 0.10 | 0.69 ± 0.03 | 4.31 ± 0.15 |
| XH | 7.65 ± 0.06 | 1.64 ± 0.14 | 0.17 ± 0.01 | 9.40 ± 0.11 | 8.54 ± 3.22 | 2.86 ± 0.07 | 0.39 ± 0.01 | 1.98 ± 0.24 |
| XG | 7.17 ± 0.18 | 1.63 ± 0.03 | 0.15 ± 0.00 | 11.03 ± 0.10 | 5.97 ± 0.84 | 2.18 ± 0.16 | 1.02 ± 0.50 | 1.43 ± 0.19 |
| HB | 7.89 ± 0.06 | 9.01 ± 0.40 | 0.79 ± 0.03 | 11.34 ± 0.04 | 2.78 ± 0.71 | 10.37 ± 0.07 | 0.95 ± 0.00 | 3.26 ± 0.81 |
| *Subsoil* | | | | | | | | |
| OB | 9.51 ± 0.12 | 0.19 ± 0.01 | 0.02 ± 0.00 | 8.25 ± 0.87 | 11.29 ± 1.26 | 1.83 ± 0.26 | 0.69 ± 0.48 | 0.39 ± 0.13 |
| DQ | 9.17 ± 0.08 | 0.78 ± 0.06 | 0.09 ± 0.01 | 8.99 ± 0.43 | 10.95 ± 1.15 | 5.33 ± 0.17 | 0.60 ± 0.03 | 4.69 ± 0.50 |
| HLHT | 8.70 ± 0.12 | 1.23 ± 0.27 | 0.13 ± 0.03 | 9.04 ± 0.26 | 11.31 ± 4.05 | 6.41 ± 0.15 | 0.57 ± 0.04 | 4.45 ± 0.38 |
| XH | 7.75 ± 0.11 | 0.53 ± 0.03 | 0.06 ± 0.00 | 8.94 ± 0.20 | 10.87 ± 2.11 | 1.88 ± 0.08 | 0.31 ± 0.01 | 1.64 ± 0.35 |
| XG | 7.79 ± 0.86 | 0.50 ± 0.03 | 0.04 ± 0.00 | 11.61 ± 0.39 | 13.07 ± 0.74 | 1.17 ± 0.13 | 0.37 ± 0.03 | 1.18 ± 0.02 |
| HB | 8.06 ± 0.06 | 1.62 ± 0.03 | 0.15 ± 0.00 | 10.73 ± 0.10 | 6.77 ± 0.16 | 11.98 ± 0.14 | 0.91 ± 0.02 | 5.73 ± 0.20 |

SOC: soil organic carbon; N: nitrogen; WEOC: water-extractable organic carbon; Fe$_d$: dithionite-extractable iron and Al$_d$: dithionite-extractable aluminum.





**Table 3. Concentrations of microbial phospholipid fatty acids (PLFAs), ratios of fungal/bacterial (F/B), Gram-positive/Gram-negative bacterial PLFAs (G+/G–), microbial metabolic quotient ($q$CO$_2$) and microbial carbon use efficiency (CUE) in the soil at the end of the incubation (mean ± standard error; n = 3).**

| Site | Control treatment | | | | | Litter-amended treatment | | | | |
|---|---|---|---|---|---|---|---|---|---|---|
| | Total PLFAs (mg g$^{-1}$ SOC) | F/B | G+/G– | $q$CO$_2$ (mg C mg$^{-1}$ PLFA day$^{-1}$) | | Total PLFAs (mg g$^{-1}$ SOC) | F/B | G+/G– | $q$CO$_2$ (mg C mg$^{-1}$ PLFA day$^{-1}$) | CUE (%) |
| | | | | | *Topsoil* | | | | | |
| OB | 0.90 ± 0.18 | 0.97 ± 0.16 | 2.45 ± 0.28 | 1.30 ± 0.30 | | 0.63 ± 0.24 | 1.27 ± 0.17 | 1.53 ± 0.46 | 2.50 ± 0.85 | 0.30 ± 0.12 |
| DQ | 0.58 ± 0.12 | 0.04 ± 0.01 | 0.78 ± 0.05 | 0.72 ± 0.09 | | 0.52 ± 0.11 | 0.04 ± 0.00 | 0.80 ± 0.04 | 0.82 ± 0.11 | 0.35 ± 0.08 |
| HLHT | 0.25 ± 0.03 | 0.04 ± 0.00 | 0.92 ± 0.04 | 1.37 ± 0.04 | | 0.33 ± 0.03 | 0.05 ± 0.00 | 1.00 ± 0.03 | 1.11 ± 0.22 | 0.47 ± 0.10 |
| XH | 0.56 ± 0.24 | 0.46 ± 0.04 | 2.82 ± 0.38 | 1.36 ± 0.51 | | 0.42 ± 0.10 | 0.38 ± 0.07 | 3.13 ± 0.67 | 1.44 ± 0.23 | 0.15 ± 0.06 |
| XG | 0.28 ± 0.03 | 0.41 ± 0.02 | 1.88 ± 0.19 | 1.24 ± 0.09 | | 0.48 ± 0.09 | 0.37 ± 0.01 | 2.20 ± 0.03 | 0.79 ± 0.15 | 0.44 ± 0.12 |
| HB | 0.06 ± 0.03 | 0.43 ± 0.00 | 0.87 ± 0.12 | 6.70 ± 3.70 | | 0.11 ± 0.02 | 0.37 ± 0.02 | 1.42 ± 0.02 | 2.47 ± 0.75 | NA |
| | | | | | *Subsoil* | | | | | |
| OB | 0.45 ± 0.09 | 2.69 ± 0.70 | 7.14 ± 4.07 | 3.37 ± 0.61 | | 0.65 ± 0.17 | 1.90 ± 0.42 | 1.72 ± 0.38 | 2.55 ± 0.71 | 0.36 |
| DQ | 0.14 ± 0.01 | 0.09 ± 0.04 | 0.90 ± 0.11 | 3.95 ± 0.16 | | 0.11 ± 0.00 | 0.05 ± 0.01 | 0.88 ± 0.01 | 5.21 ± 0.23 | 0.31 ± 0.09 |
| HLHT | 0.16 ± 0.07 | 1.72 ± 1.58 | 1.06 ± 0.11 | 6.92 ± 4.02 | | 0.30 ± 0.04 | 0.07 ± 0.01 | 0.86 ± 0.05 | 2.32 ± 0.44 | 0.28 ± 0.02 |
| XH | 0.41 ± 0.06 | 0.19 ± 0.02 | 2.48 ± 0.14 | 0.89 ± 0.12 | | 0.40 ± 0.14 | 0.22 ± 0.01 | 3.86 ± 0.25 | 1.15 ± 0.57 | 0.28 ± 0.19 |
| XG | 0.38 ± 0.05 | 0.35 ± 0.03 | 2.20 ± 0.04 | 0.98 ± 0.11 | | 0.33 ± 0.05 | 0.35 ± 0.01 | 2.30 ± 0.03 | 1.01 ± 0.12 | 0.20 |
| HB | 0.13 ± 0.01 | 0.29 ± 0.01 | 1.31 ± 0.08 | 4.34 ± 0.55 | | 0.14 ± 0.02 | 0.27 ± 0.02 | 1.69 ± 0.20 | 4.69 ± 0.77 | NA |



**Table 4. Specific activity of oxidase (mM g$^{-1}$ SOC h$^{-1}$) and hydrolases (μM g$^{-1}$ SOC h$^{-1}$) in the soil at the end of the incubation (mean ± standard error; n = 3).**

| Site | Control treatment | | | | | Litter-amended treatment | | | | |
|------|--------|----|----|----|-----|--------|----|----|----|-----|
| | Oxidase | Hydrolases | | | | Oxidase | Hydrolases | | | |
| | PO | AG | BG | AP | LAP | PO | AG | BG | AP | LAP |
| | | | | | *Topsoil* | | | | | |
| OB | 15.86 ± 3.78 | 0.02 ± 0.00 | 0.05 ± 0.02 | 0.07 ± 0.01 | 0.64 ± 0.09 | 16.60 ± 0.82 | 0.01 ± 0.00 | 0.04 ± 0.01 | 0.08 ± 0.01 | 0.52 ± 0.06 |
| DQ | 6.92 ± 0.84 | 0.34 ± 0.10 | 2.99 ± 0.73 | 9.93 ± 2.43 | 6.56 ± 0.25 | 4.49 ± 0.12 | 0.19 ± 0.03 | 2.52 ± 0.52 | 5.16 ± 0.46 | 6.25 ± 0.85 |
| HLHT | 2.52 ± 0.23 | 0.08 ± 0.02 | 2.06 ± 0.26 | 4.00 ± 1.18 | 3.21 ± 0.65 | 2.39 ± 0.48 | 0.03 ± 0.00 | 0.69 ± 0.07 | 1.09 ± 0.52 | 2.39 ± 0.15 |
| XH | 3.26 ± 0.43 | 0.09 ± 0.05 | 1.02 ± 0.40 | 2.12 ± 0.27 | 0.28 ± 0.03 | 2.84 ± 0.78 | 0.08 ± 0.06 | 1.33 ± 0.26 | 2.52 ± 0.35 | 0.36 ± 0.04 |
| XG | 1.65 ± 0.15 | 0.20 ± 0.03 | 1.86 ± 0.28 | 2.30 ± 0.21 | 0.18 ± 0.00 | 1.39 ± 0.55 | 0.20 ± 0.05 | 2.00 ± 0.43 | 2.58 ± 0.64 | 0.21 ± 0.04 |
| HB | 0.46 ± 0.09 | 0.01 ± 0.00 | 0.03 ± 0.01 | 0.03 ± 0.00 | 0.16 ± 0.07 | 0.48 ± 0.09 | 0.00 ± 0.00 | 0.01 ± 0.00 | 0.02 ± 0.01 | 0.14 ± 0.08 |
| | | | | | *Subsoil* | | | | | |
| OB | 26.22 ± 2.90 | 0.02 ± 0.01 | 0.05 ± 0.01 | 0.08 ± 0.03 | 1.24 ± 0.47 | 24.63 ± 3.38 | 0.02 ± 0.00 | 0.04 ± 0.01 | 0.09 ± 0.01 | 0.84 ± 0.09 |
| DQ | 12.87 ± 2.32 | 0.00 ± 0.00 | 0.72 ± 0.15 | 14.19 ± 1.27 | 23.18 ± 0.55 | 13.13 ± 2.59 | 0.26 ± 0.10 | 1.19 ± 0.10 | 16.75 ± 2.17 | 21.39 ± 1.91 |
| HLHT | 10.43 ± 1.49 | 0.15 ± 0.07 | 1.48 ± 0.33 | 17.36 ± 4.27 | 17.65 ± 4.74 | 5.45 ± 2.29 | 0.67 ± 0.23 | 2.11 ± 0.58 | 20.38 ± 2.43 | 21.62 ± 5.59 |
| XH | 4.03 ± 0.60 | 0.08 ± 0.03 | 0.89 ± 0.11 | 1.03 ± 0.21 | 0.16 ± 0.02 | 4.84 ± 1.17 | 0.03 ± 0.01 | 0.89 ± 0.22 | 1.09 ± 0.17 | 0.17 ± 0.04 |
| XG | 1.14 ± 0.35 | 0.25 ± 0.08 | 3.22 ± 0.55 | 5.12 ± 1.20 | 0.41 ± 0.03 | 1.99 ± 0.00 | 0.36 ± 0.02 | 3.22 ± 0.07 | 4.60 ± 0.43 | 0.28 ± 0.03 |
| HB | 6.10 ± 1.35 | 0.15 ± 0.06 | 0.47 ± 0.29 | 0.63 ± 0.36 | 6.72 ± 2.51 | 5.17 ± 0.64 | 0.14 ± 0.06 | 0.51 ± 0.17 | 0.68 ± 0.36 | 8.91 ± 1.94 |

PO: phenol oxidase; AG: α-glucosidase; BG: β-glucosidase; AP: alkaline phosphatase; LAP: leucine-aminopeptidase.





**Table 5. Standardized partial regression coefficients of the multiple stepwise regression analysis for substrate mineralization potentials with environmental variables.**

| Mineralization potential[1] | Environmental variables | | | | | | | $R^2$ | $p$ |
|---|---|---|---|---|---|---|---|---|---|
| | Aridity index | Soil minerals[2] | Soil pH | SOM property[3] | PLFAs[4] | Hydrolases[5] | Phenol oxidase | | |
| $R_{control}$ | ns | ns | ns | ns | ns | ns | **0.92** | 0.85 | < 0.01 |
| $R_{SOC}$ | ns | ns | ns | ns | ns | ns | **0.87** | 0.76 | < 0.01 |
| $R_{litter}$ | ns | ns | ns | ns | **0.65** | ns | ns | 0.42 | < 0.01 |

[1]$R_{control}$ and $R_{SOC}$ refer to the mineralization potential of soil organic carbon (SOC) in the control and leaf-amended treatments, respectively while $R_{litter}$ refers to the mineralization potential of leaf litter; [2]soil minerals are represented by the first principal component in the principal component analysis involving $Fe_d$, $Al_d$ and clay; [3]soil organic matter (SOM) property is represented by the first principal component in the principal component analysis involving SOC, SOC:N ratio and water-extractable organic carbon (WEOC) contents; [4]PLFAs: phospholipid fatty acids; [5]hydrolases are represented by the first principal component in the principal component analysis involving α-glucosidase, β-glucosidase, alkaline phosphatase and leucine-aminopeptidase; ns: not significant. Bold fonts correspond to the highest coefficient values and hence the strongest influence by the corresponding environmental variable.





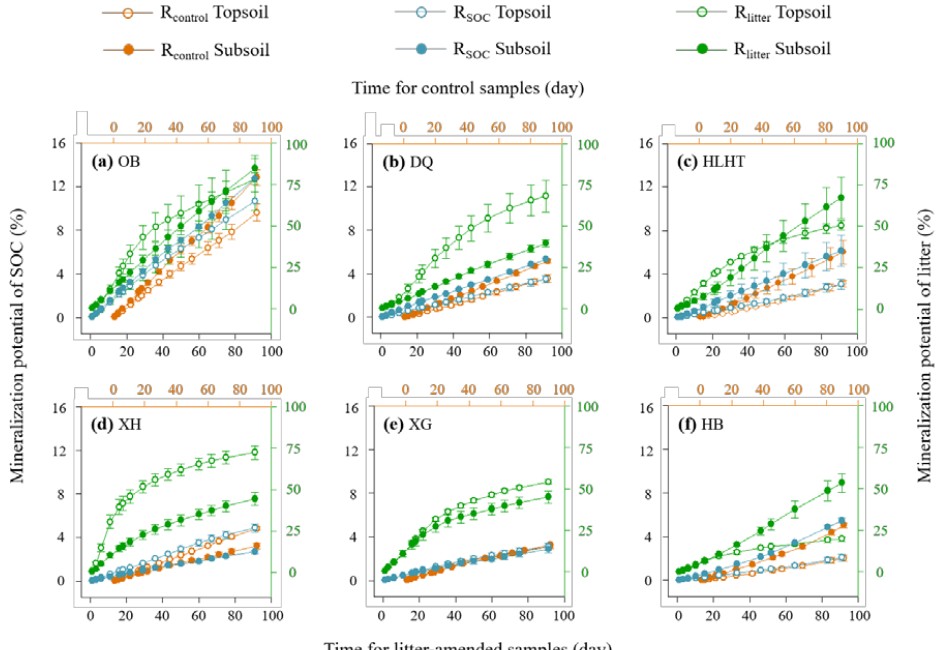

**Figure 1. Mineralization potential of soil organic carbon (SOC) and leaf litter during the 91-day incubation. R$_{control}$ and R$_{SOC}$ refer to the mineralization potential of SOC in the control and leaf-amended treatments, respectively while R$_{litter}$ refers to the mineralization potential of leaf litter. Mean values are shown with standard error (n = 3). The x axis (top) for R$_{control}$ is shifted to the right relative to the leaf-amended treatments (bottom axis) for better illustration.**



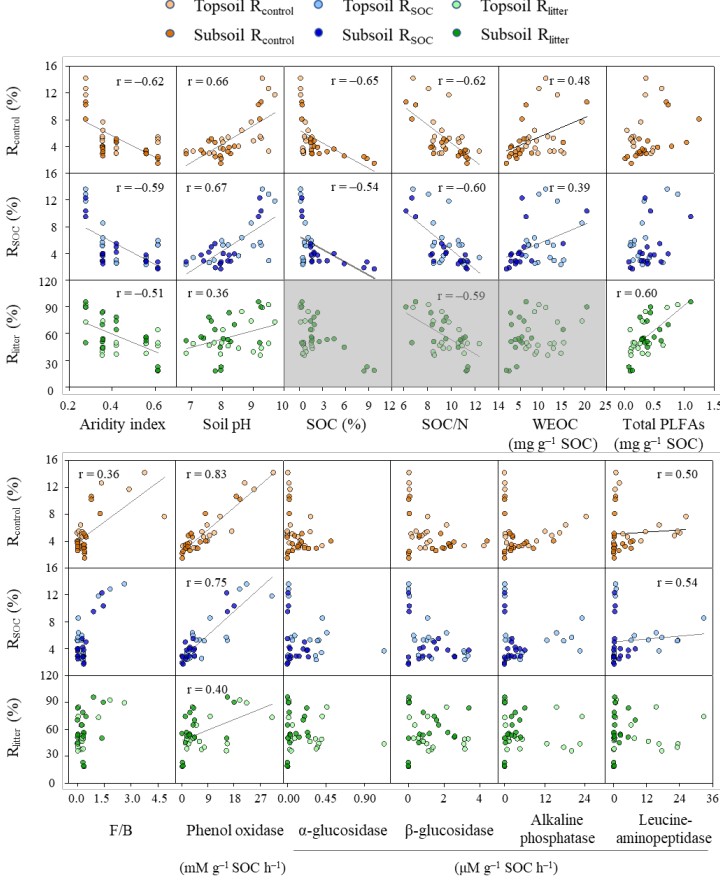

**Figure 2. Relationships between the mineralization potential of substrates and key environmental variables.** $R_{control}$ and $R_{SOC}$ refer to the mineralization potential of soil organic carbon (SOC) in the control and leaf-amended treatments, respectively while $R_{litter}$ refers to the mineralization potential of leaf litter. N: nitrogen; WEOC: water-extractable organic carbon; F/B: ratios of fungal to bacterial PLFAs; PLFAs: phospholipid fatty acids; Field replicates are shown as individual data points. Black lines represent significant Spearman correlations for non-normally distributed data or Pearson correlations for normally distributed data (i.e., between pH and $R_{litter}$; $p < 0.05$). Three boxes of $R_{litter}$ are shaded in grey because the examined soil properties do not describe litter quality and hence should not be correlated.





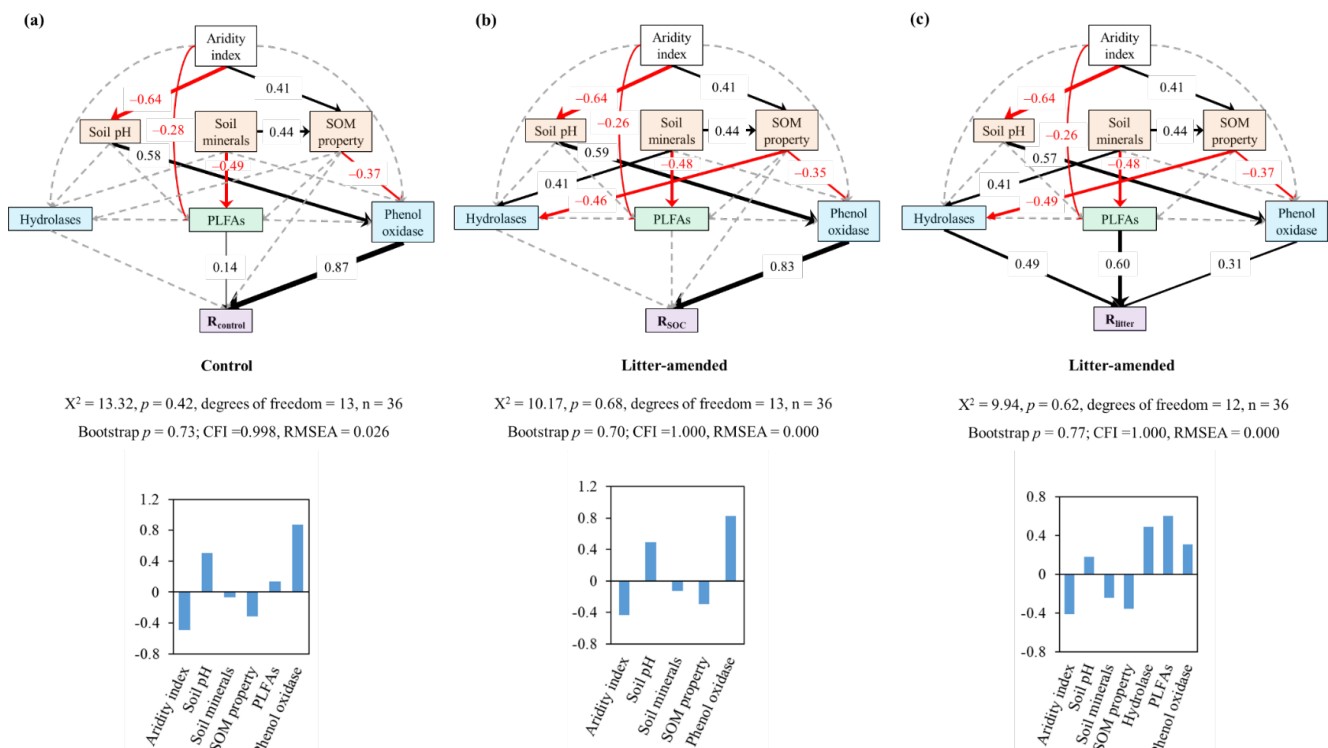

**Figure 3. Best-supported structural equation models (SEMs) disentangling cascading effects of environmental variables on substrate mineralization. R<sub>control</sub> (a) and R<sub>SOC</sub> (b) refer to the mineralization potential of soil organic carbon (SOC) in the control and leaf-amended treatments, respectively, while R<sub>litter</sub> (c) refers to the mineralization potential of leaf litter. Black and red arrows indicate positive and negative flows of causality ($p < 0.05$), respectively. Grey dotted lines indicate insignificant pathways from priori models (Figure S2). Numbers on the arrow indicate significant standardized path coefficients, proportional to the arrow width. Environmental variables are categorized into climate (i.e., aridity index), extracellular enzymes (in blue) including hydrolyses and phenol oxidase, microbial biomass (in green) represented by phospholipid fatty acids (PLFAs) and soil properties (in orange) including soil minerals, soil pH and soil organic matter (SOM) property. Soil minerals, SOM property and hydrolases are defined by a principle component analysis (Table S5).**