# Peer review of "Past aridity's effect on carbon mineralization potentials in grassland soils"

_Biogeosciences, 2019_

## Referee Comment (RC1) · Tida Ge (Referee) · 29 May 2019

This is a nice work to quantitatively assess mechanisms contributing to the effect of past aridity on mineralization potentials.

---

## Referee Comment (RC2) · Anonymous Referee #2 · 16 Jul 2019

The manuscript bg-2019-167 reported results from a 91-day soil incubation experiment using top and deep soils from six grassland sites in China. The authors showed different controls of native SOC and added litter decomposition (i.e. potential mineralization rate) by different microbial variables (biomass vs. enzyme) by statistical analyses (regression and SEM). Overall, the topic is relevant to Biogeosciences, the writing is clear, and the analysis and interpretation are mostly robust and reasonable. I have a few suggestions or concerns for the authors to consider in revision.

1. A key point in the original design is the comparison between top vs. deep soils. For example, how the potential decomposition rate of native SOC vs. added litter and the priming effect of litter on native SOC differ between top vs. deep soils in these six sites. However, the analysis and comparison between soil layers are not adequate

enough. I suggest the authors to show more direct results (mixed-effects models) on this comparison, and analyze data (e.g. regression, SEM) separately for top vs. deep soils.

2. The results of priming effect were not shown. Even if they were statistically neutral, some illustrations (figures or tables) and analyses (mixed-effects model, regression) can be helpful for readers to understand how the priming effect vary with depth and site (and what are the driving variables of PE across these depth intervals and sites).

3. The depth intervals for deep soils were not consistent across sties. Some explanation and justification is needed.

4. More information on the 13C-labeled litter is needed. How were they labeled? Were they uniformly labeled (with data to support this)?

5. The rate of litter addition need more explanation and justification. What criteria was used? How were these rates determined? What were the rates (gram litter C) per gram soil, per gram SOC, and per gram MBC for all 12 soils?

6. PLFA, normally the unit is nmol. How did you go from nmol to mg? A table with all detected markers and their assigned groups used in this study would be helpful.

7. CUE, the value was extremely low, because of the method used in this study (91-day incubation, conversion from PLFA to biomass C). Probably, it is more appropriate to use another term for this study.

8. Enzyme activities were sensitive to the pH of buffer. As different soils had different pH, did you control buffer pH for each soil?

Specific comments

Table 5: the results (most were n.s.) were a little surprising. Do results change if you analyze the two soil layers separately?

Figure 1: what were the results of priming effect?

Figure 2 and 3: Do results change if you analyze the two soil layers separately?

---

## Author Comment (AC1) · 7 Aug 2019

**Response to Referee #2**

We appreciate Referee #2's critical and detailed assessment of our manuscript and we are grateful for his/her constructive comments which helped us to greatly refine our paper. Here we provide a point-to-point response to all the issues raised by the referee. Our response (in blue font) follows the original comment (in black font). The revised version with changes were highlighted in red. We hope our replies and revisions will satisfy all the requests.

**Comment 1:**

A key point in the original design is the comparison between top vs. deep soils. For example, how the potential decomposition rate of native SOC vs. added litter and the priming effect of litter on native SOC differ between top vs. deep soils in these six sites. However, the analysis and comparison between soil layers are not adequate enough. I suggest the authors to show more direct results (mixed-effects models) on this comparison, and analyze data (e.g. regression, SEM) separately for top vs. deep soils.

Thank you for this comment. We did compare top- vs. subsoil differences. However, there were no significant differences ($p > 0.05$) in the SOC decomposition rates (i.e., $R_{control}$ and $R_{SOC}$) between top- and subsoils except HB having higher $R_{control}$ in the sub- than topsoil ($p < 0.05$). Litter decomposition rare ($R_{litter}$) were also similar in the top- and subsoils except that XH and HB showed higher values in top- and subsoils than their counterparts, respectively ($p < 0.05$). None of the soils showed any priming effect in this study (please see detailed information in Table R1 and now added in Section 3.3). Hence, comparisons between top- and subsoils were not emphasized here.

**Table R1.** Mineralization potentials of organic carbon in top- and subsoils of this study. $R_{control}$ and $R_{SOC}$ refer to the mineralization potential of soil organic carbon (SOC) in the control and litter-amended treatments, respectively while $R_{litter}$ refers to the mineralization potential of litter.

| Site | $R_{control}$ (%) | $R_{SOC}$ (%) | $R_{litter}$ (%) |
|------|------------------|---------------|------------------|
| | | *Topsoil* | |
| OB | 9.63 ± 0.79 | 10.69 ± 0.83 | 78.31 ± 14.44 |
| DQ | 3.57 ± 0.34 | 3.48 ± 0.34 | 68.02 ± 9.41 |
| HLHT | 3.11 ± 0.30 | 3.08 ± 0.45 | 50.56 ± 3.02 |
| XH | 4.82 ± 0.15 | 4.85 ± 0.37 | 72.22 ± 4.15 |
| XG | 3.09 ± 0.17 | 3.10 ± 0.32 | 54.24 ± 1.23 |
| HB | 2.01 ± 0.30 | 2.06 ± 0.32 | 19.69 ± 1.42 |
| | | *Subsoil* | |
| OB | 12.83 ± 0.73 | 12.74 ± 0.52 | 84.80 ± 6.01 |
| DQ | 5.15 ± 0.18 | 5.31 ± 0.16 | 39.50 ± 2.28 |
| HLHT | 6.00 ± 1.07 | 6.15 ± 1.42 | 67.23 ± 12.33 |
| XH | 3.21 ± 0.26 | 2.69 ± 0.15 | 44.48 ± 3.86 |
| XG | 3.28 ± 0.01 | 2.85 ± 0.30 | 45.05 ± 3.64 |
| HB | 5.01 ± 0.22 | 5.46 ± 0.22 | 53.30 ± 5.59 |

Nevertheless, we tried to use mixed-effect models and SEM to analyze the main influencing

factors for organic carbon mineralization potentials at different depths separately, yet in vain due to inadequate number of samples. Multiple stepwise regression is hence utilized for separate depth and we have added the corresponding results in Section 3.5 as follows (please see revised Table 5 for the detailed data):

*"Furthermore, when the top- and subsoils are considered separately, phenol oxidase activities remain the only most important regulator for SOC mineralization at both depths. PLFAs remain the only important regulator for $R_{litter}$ in the subsoil while PLFAs as well as aridity index govern $R_{litter}$ in the topsoil."*

Hence, our results and conclusions are consistent for both depths of soils.

**Comment 2**

The results of priming effect were not shown. Even if they were statistically neutral, some illustrations (figures or tables) and analyses (mixed-effects model, regression) can be helpful for readers to understand how the priming effect vary with depth and site (and what are the driving variables of PE across these depth intervals and sites).

Following your advice, we have added a figure (Fig. S4) to exhibit the comparison of mineralization potentials of soil organic carbon (SOC) in the control ($R_{control}$) and leaf-amended treatments ($R_{SOC}$). We use Paired T (for normal distributed data) or Wilcoxon to test whether there is priming effect at different site. The results show no priming effect for any of the studied soil ($p > 0.05$). Hence, none of the soils showed any priming effect (this is now added in Section 3.3).

**Comment 3**

The depth intervals for deep soils were not consistent across sties. Some explanation and justification is needed.

Soils from two horizons were collected with varying depths for the subsoil due to different development and depths of soil profiles at varied sites. This is now added into Section 2.1.

**Comment 4**

More information on the $^{13}$C-labeled litter is needed. How were they labeled? Were they uniformly labeled (with data to support this)?

The grass leaves were purchased from commercial suppliers and continuously labelled with $CO_2$ gas with 99.9 atom% $^{13}$C for 3 months in a growth chamber. This information is now added in Section 2.3.

**Comment 5**

The rate of litter addition need more explanation and justification. What criteria was used? How were these rates determined? What were the rates (gram litter C) per gram soil, per gram SOC, and per gram MBC for all 12 soils?

According to Blagodatskaya and Kuzyakov (2008), SOC priming is expected when the amendment rate of substrate C is lower than 50% of soil microbial biomass carbon (MBC). Given that MBC generally makes up 1–4% of total SOC (Anderson and Domsch, 1989; Sparling, 1992), we intended to add similar amount of litter C corresponding to 0.7% SOC for the respective soils to induce priming in our original design. However, as we admitted in our paper, due to an oversight in calculation, the second batch of soils received less external C (0.29% of SOC). Nonetheless, no priming was induced in either batch of incubation, likely because we used litter instead of labile carbon such as glucose in our experiment. For your information, detailed amendment rates for each soil are listed in Table R2.

**Table R2. Rates of litter addition for all the studied soils.**

| Site | mg C g$^{-1}$ soil | mg C g$^{-1}$ SOC | mg C g$^{-1}$ soil | mg C g$^{-1}$ SOC |
|---|---|---|---|---|
| | *Topsoil* | | *Subsoil* | |
| OB | 0.01 | 2.88 | 0.01 | 2.88 |
| DQ | 0.07 | 2.88 | 0.02 | 2.88 |
| HLHT | 0.13 | 2.88 | 0.04 | 2.88 |
| XH | 0.05 | 2.88 | 0.02 | 2.88 |
| XG | 0.05 | 2.88 | 0.01 | 2.88 |
| HB | 0.63 | 7.00 | 0.11 | 7.00 |

References:

Anderson, T-H. and Domsch, K. H.: Ratios of microbial biomass carbon to total organic carbon in arable soils, Soil Biol. Biochem., 21, 471–479, 1989.

Blagodatskaya, E. and Kuzyakov, Y.: Mechanisms of real and apparent priming effects and their dependence on soil microbial biomass and community structure: critical review, Biol. Fert. Soils, 45, 115–131, 2008.

Sparling, G. P.: Ratio of microbial biomass carbon to soil organic carbon as a sensitive indicator of changes in soil organic matter, Aust. J. Soil Res., 30, 195–207, 1992.

**Comment 6**

PLFA, normally the unit is nmol. How did you go from nmol to mg? A table with all detected markers and their assigned groups used in this study would be helpful.

For the units of PLFAs, mg is also frequently used (Feng et al., 2007; Jin and Evans, 2010; Hopkins et al., 2014; Wang et al., 2017) other than nmol. Quantification of individual PLFA was achieved by comparison with the internal standard in the total ion current assuming similar response factor for various FAMEs (in the units of either mg or umol). Given the relatively similar molecular mass of target PLFAs (typically having 14-18 carbon atoms), conversion between the two units does not lead to much difference in the results. In this study, as we need to calculate CUE' based on the C content of PLFAs, we choose to use the

units of mg.

In addition, following your advice, we have also added a table (Table S6) to exhibit the detected markers and their assigned groups.

References:

Feng, X., Nielsen, L. L. and Simpson, M. J.: Responses of soil organic matter and microorganisms to freeze–thaw cycles, Soil Biol. Biochem., 39, 2027–2037, 2007.

Hopkins, F. M., Filley, T. R., Gleixner, G., Lange, M., Top, S. M. and Trumbore, S. E.: Increased belowground carbon inputs and warming promote loss of soil organic carbon through complementary microbial responses, Soil Biol. Biochem., 76, 57–69, 2014.

Jin, V. L. and Evans, R. D.: Microbial $^{13}$C utilization patterns via stable isotope probing of phospholipid biomarkers in Mojave Desert soils exposed to ambient and elevated atmospheric $CO_2$, Glob. Change Biol., 16, 2334–2344, 2010.

Wang, J., Liu, L., Wang, X., Yang, S., Zhang, B., Li, P., Qiao, C., Deng, M., Liu, W. and Treseder, K.: High night-time humidity and dissolved organic carbon content support rapid decomposition of standing litter in a semi-arid landscape, Funct. Ecol., 31, 1659–1668, 2017.

**Comment 7**

CUE, the value was extremely low, because of the method used in this study (91- day incubation, conversion from PLFA to biomass C). Probably, it is more appropriate to use another term for this study.

Thank you for this suggestion. We have replaced CUE with CUE' to differentiate PLFA (instead of microbial biomass carbon)-based CUE in the manuscript.

**Comment 8**

Enzyme activities were sensitive to the pH of buffer. As different soils had different pH, did you control buffer pH for each soil?

Yes, we did buffer pH for enzyme assay. Actually, all soils in this experiment were alkaline. We tested the pH of the soil solution, which was in the range of the buffer. Hence the pH was consistent for different soils during enzyme assay in this study.

**Specific comments**
Table 5: the results (most were n.s.) were a little surprising. Do results change if you analyze the two soil layers separately?

The main results remain unchanged, except that both aridity index and PLFAs influence $R_{litter}$ of the topsoils when the two soil layers are analyzed separately. The statistical results

are now added in Table 5 and mentioned in Section 3.5.

Figure 1: what were the results of priming effect?

There was no priming effect in our study. Figure S4 is added to show the results.

Figure 2 and 3: Do results change if you analyze the two soil layers separately?

When the two soil layers are separated, some of the correlations in Figure 2 are not significant due to the relatively smaller range of variables and SEM cannot be conducted due to inadequate number of samples. As we mentioned earlier, there are no significant or consistent differences in the decomposition rates between top- and subsoils and comparisons between depths are hence not emphasized in this study. Furthermore, multiple stepwise regression analysis shows similar results when the topsoil and subsoil are separated or combined (Table 5). Hence, we choose to combine both soil layers for the statistical analysis to reveal a more universal mechanism regulating organic carbon mineralization potentials and to produce more reliable SEM results.

---

## Author Comment (AC2) · 7 Aug 2019

We appreciate Referee #1's positive assessment of our work!

---

## Author Response (AR1)

**Response to Editor on bg-2019-167**

**Editor's comments:**

Comments to the Author:

The ms has been thouroughly improved by the authors and can be considered to be publish in BG. there are still two main issues I am concerning as following:

We sincerely thank the editor's supportive comments. As detailed below, we have modified our manuscript accordingly.

**Comment 1**

Up to 95% of litter was mineralized in such a short period of incubation, this needs to be clarified.

Thank you for this comment. We have added the corresponding clarification in Section 4.1 as below:

*"Furthermore, the $R_{litter}$ values measured under optimal conditions in this study (25˚C; soil moisture: 55–60% of water holding capacity) are comparable to the litter mineralization rates reported in "real-world" conditions such as in field litterbag experiments. For instance, Wang et al. (2014) reported that >70% of E. speciosus litter degraded under warm (12–35˚C) and humid conditions within 90 days. Shaw and Harte (2001) found that nearly 73% of forb litter was lost within 46 days in a subalpine meadow. Sievers et al. (2018) also discovered that the litter of hairy vetch and cereal rye degraded by 90% in cropland within 84 days. Hence, we consider the $R_{litter}$ measured in this study to reflect optimal decomposition rates of litter in semiarid regions."*

References:

Shaw, M. R., and Harte, J.: Control of litter decomposition in a subalpine meadow-sagebrush steppe ecotone under climate change, Ecol. Appl., 11, 1206–1223, 2001.

Sievers, T., and Cook, R.L.: Aboveground and root decomposition of cereal rye and hairy vetch cover crops, Soil Sci. Soc. Am. J., 82, 147–155, 2018.

Wang, G., Zhang, L., Zhang, X., Wang, Y., and Xu, Y.: Chemical and carbon isotopic dynamics of grass organic matter during litter decompositions: A litterbag experiment, Org. Geochem., 69, 106–113, 2014.

**Comment 2**

Soil PE and other C processes (litter decompose) were not strongly linked with neither edaphic abiological variables, pH, mineral, for example, nor microbial biomass and community, what else potential mechanisms might be?

This is a very good point! PE was not detected in our study. Its regulating factors are complex and beyond the scope of our paper. Litter decomposition rates, on the other hand, are most strongly influenced by PLFA abundances in our incubation experiment (shown in Table 5). However, PLFAs, along with other measured variables, only explained 42% of the $R_{litter}$ variance ($R^2 = 0.42$). Hence, there are still other factors regulating $R_{litter}$ that are not depicted by our analysis. The potential influencing mechanisms include (but are not limited to) radical attack by reactive oxygen species that are widely observed in natural soils and protection by soil aggregation. These considerations are now added to the final part of Section 4.3.